# A systematic approach to the development of a safe live attenuated Zika vaccine

Swee Sen Kwek[1], Satoru Watanabe[1], Kuan Rong Chan[1], Eugenia Z. Ong[1], Hwee Cheng Tan[1], Wy Ching Ng[1], Mien T.X. Nguyen[2], Esther S. Gan[1], Summer L. Zhang[1], Kitti W.K. Chan [1,3], Jun Hao Tan[1], October M. Sessions[1], Menchie Manuel[1], Julien Pompon[1,4], Camillus Chua [5], Sharifah Hazirah[5], Karl Tryggvason[2,6], Subhash G. Vasudevan[1,3] & Eng Eong Ooi[1,3,7,8]

Zika virus (ZIKV) is a flavivirus that can cause congenital disease and requires development of an effective long-term preventative strategy. A replicative ZIKV vaccine with properties similar to the yellow fever 17D (YF17D) live-attenuated vaccine (LAV) would be advantageous, as a single dose of YF17D produces lifelong immunity. However, a replicative ZIKV vaccine must also be safe from causing persistent organ infections. Here we report an approach to ZIKV LAV development. We identify a ZIKV variant that produces small plaques due to interferon (IFN)-restricted viral propagation and displays attenuated infection of endothelial cells. We show that these properties collectively reduce the risk of organ infections and vertical transmission in a mouse model but remain sufficiently immunogenic to prevent wild-type ZIKV infection. Our findings suggest a strategy for the development of a safe but efficacious ZIKV LAV.

---

[1] Programme in Emerging Infectious Diseases, Duke-National University of Singapore Medical School, Singapore 169857, Singapore. [2] Programme in Cardiovascular and Metabolic Disorders, Duke-National University of Singapore Medical School, Singapore 169857, Singapore. [3] Department of Microbiology and Immunology, Yong Loo Lin School of Medicine, National University of Singapore, Singapore 117597, Singapore. [4] MIVEGEC, IRD, CNRS, Univ. Montpellier, 34394 Montpellier, France. [5] SingHealth Translational Immunology and Inflammation Centre, SingHealth, Singapore 169856, Singapore. [6] Department of Medical Biochemistry and Biophysics, Division of Matrix Biology, Karolinska Institute, 17177 Stockholm, Sweden. [7] Saw Swee Hock School of Public Health, National University of Singapore, Singapore 117549, Singapore. [8] Singapore-MIT Alliance for Research and Technology, Infectious Diseases Interdisciplinary Group, Singapore 138602, Singapore. These authors contributed equally: Satoru Watanabe, Kuan Rong Chan. Correspondence and requests for materials should be addressed to E.E.O. (email: engeong.ooi@duke-nus.edu.sg)

The emergence of Zika virus (ZIKV) as a cause of fetal malformations in infected expectant women has caused considerable alarm[1]. These malformations include severe microcephaly and blindness, which are lifelong disabilities that impose significant burden on the victims, their families, and their societies. Preventing such congenital infection and hence malformation is thus important if we are to reduce the global Zika disease burden[2]. In that context, we could draw epidemiological lessons from the success story of a childhood vaccination program in preventing congenital rubella disease. The long-lasting immunity engendered by the RA 27/3 rubella live-attenuated vaccine (LAV) and its use in childhood vaccination programs resulted in high herd immunity levels that prevented maternal rubella and consequent congenital rubella disease[3–5]. Although there are differences in the mode of transmission and transmission potential of ZIKV and rubella virus, achieving long-lasting high population immunity levels through vaccination could prove effective in preventing congenital Zika syndrome as it did for congenital rubella.

Various ZIKV vaccines have been reported and are mostly directed at inducing antibodies against the envelope of the virus, with the exception of a recently reported vaccine candidate that targets the non-structural protein NS1[6,7]. However, whether these vaccine constructs can elicit long-term immunity remains to be determined. Indeed, long-lasting immunity may be the critical selection factor for the Zika vaccine construct that would eventually be deployed, given that years of low-level ZIKV transmission may intersperse Zika epidemics[1,8]. Moreover, long-lasting immunity would also address the theoretical concern that sub-neutralizing levels of antibodies could paradoxically enhance ZIKV infection[9]. Consequently, the target product profile of a Zika vaccine may need to be very similar to that of the yellow fever 17D (YF17D) LAV strain, where a single dose confers lifelong immunity against YF virus[10]. Indeed, observations from decades of experience with vaccine efficacy suggest that LAVs in general produce the longest duration of immunity compared to other vaccine constructs. However, studies to demonstrate long-lasting immunity are both costly and time-consuming, which are formidable barriers against clinical translation. Identifying a molecular basis for selecting ZIKV vaccine candidate(s) with YF17D-like properties could thus enable a pragmatic approach to the development of a safe and immunogenic ZIKV LAV.

The molecular underpinnings of the product profile of YF LAV has been a subject of research interest. This replicative vaccine induces robust B- and T-cell responses[11–13]. Indeed, an important lesson that is being drawn from dengue vaccine development is the need for robust CD8+ T-cell memory[14,15]. Moreover, we have recently shown that a brief pulse of antigen was not useful immunogenically. Instead, an important determinant of robust neutralizing antibody response is the duration of viremia; viremia at day 7 but not day 3 post-YF LAV vaccination directly correlated with eventual YF neutralizing antibody titer[16]. This viremia also drives type I interferon (IFN) as well as other innate immune responses that shape the development of adaptive immunity[16]. Consequently, a ZIKV vaccine that elicits long-lasting immunity would need to recapitulate these features of YF LAV. However, an infectious ZIKV vaccine poses safety concerns, critical among which is the predisposition of ZIKV to cause persistent infection that results in either serious disease or person-to-person transmission through sexual contact[17]. Defining a pathway to identify ZIKV strains attenuated in both acute disease and persistent infection would thus be important to pave a strategy for the development of an effective and safe ZIKV LAV. We suggest here that in addition to the attenuating properties previously identified for YF17D and DENV2 vaccine strain PDK53 (a flavivirus LAV that has successfully completed phase 1 and 2 clinical trials[7] as

well as the component and backbone of Takeda's DENVax formulation), a ZIKV strain attenuated in endothelial cell (EC) infection may reduce the risk of persistent infection in vital organs for adequate safety.

Here, we isolated and, using an infectious clone, rescued a small-plaque variant of a French polynesian ZIKV isolate, DN-2. Using DN-2, we show that screening for the ability to induce similar gene expression as YF17D in antigen-presenting cells, while having reduced EC infectivity could enable a systematic approach to developing a safe and immunogenic ZIKV LAV.

## Results

**Identification of small-plaque ZIKV variant**. A small-plaque phenotype was the primary selection criteria to select viral strains for additional serial passage in cells for further attenuation and vaccine development. This process, however, is lengthy. To accelerate the identification of such ZIKV LAV candidates, we took advantage of the inherent error-prone property of flaviviral RNA-dependent RNA polymerase[18]. We posited that even a few passages of wild-type ZIKV in cell cultures would be sufficient to generate a small proportion of genetic variants with IFN-restricted propagative fitness similar to YF17D and PDK53. We thus expanded the French polynesian strain, PF13/251013-18 (KX369547, hereon referred to as PF13) four times in Vero cells and once in C6/36 cells, the latter having previously been shown to diversify the closely related DENV genome in culture[19]. C6/36-derived PF13 was then plaqued on BHK-21 cells. Consistent with our hypothesis, we observed heterogeneity in plaque sizes of PF13 (Fig. 1a). We next picked the cells at the edge of small plaques and extracted the RNA for full viral genome sequencing.

This approach identified four different genetic variants of ZIKV, which we named DN-1 to −4; the amino acid changes relative to the consensus PF13 sequence are shown in Fig. 1a (non-coding changes in Supplementary Fig. 1). Infectious clones of PF13 and DN-1 to −4 were synthesized and the viruses rescued on Vero cells. DN-2 produced the smallest plaques in a Vero cell plaque assay (Fig. 1a). Moreover, the plaque size of DN-2 was significantly smaller than DN-1 even though these two strains differ only by an alanine-to-guanine substitution at genome position 948 that corresponds to a methionine to valine amino acid change in the membrane (M) gene. This M protein change is unique among both wild-type mosquito-borne flaviviruses and their LAV strains (Fig. 1b).

Using next-generation sequencing (NGS), we found that this A948G mutation was also stable up to seven passages of DN-2 on Vero cells (Table 1). Moreover, DN-2 showed fewer single-nucleotide variants during serial passaging than DN-1 (Fig. 2). However, DN-2 did gain a mutation in the NS1 gene at nucleotide position 2904 at the fourth passage that resulted in a threonine-to-serine amino acid substitution (Table 1). This mutation is likely an adaptation to Vero cell as we have derived DN-2 through minimal number of in vitro passages. We do not know the clinical significance of this mutation, which is in the wing domain of NS1[20,21] and has not been associated with altered flaviviral fitness. As the stability of DN-2 genome will need to be evaluated in whatever cell line that is selected to produce this virus for further clinical evaluation, we explored the phenotype of DN-2 that has been expanded fewer than three passages in Vero cells.

**IRF3-restricted plaque size**. We have previously shown that both YF17D and PDK53 induce robust type I IFN responses upon infection that limited their ability to propagate in a cell monolayer, thus forming small plaques in a plaque assay[22]. This is an important determinant of a small-plaque phenotype since a slow-

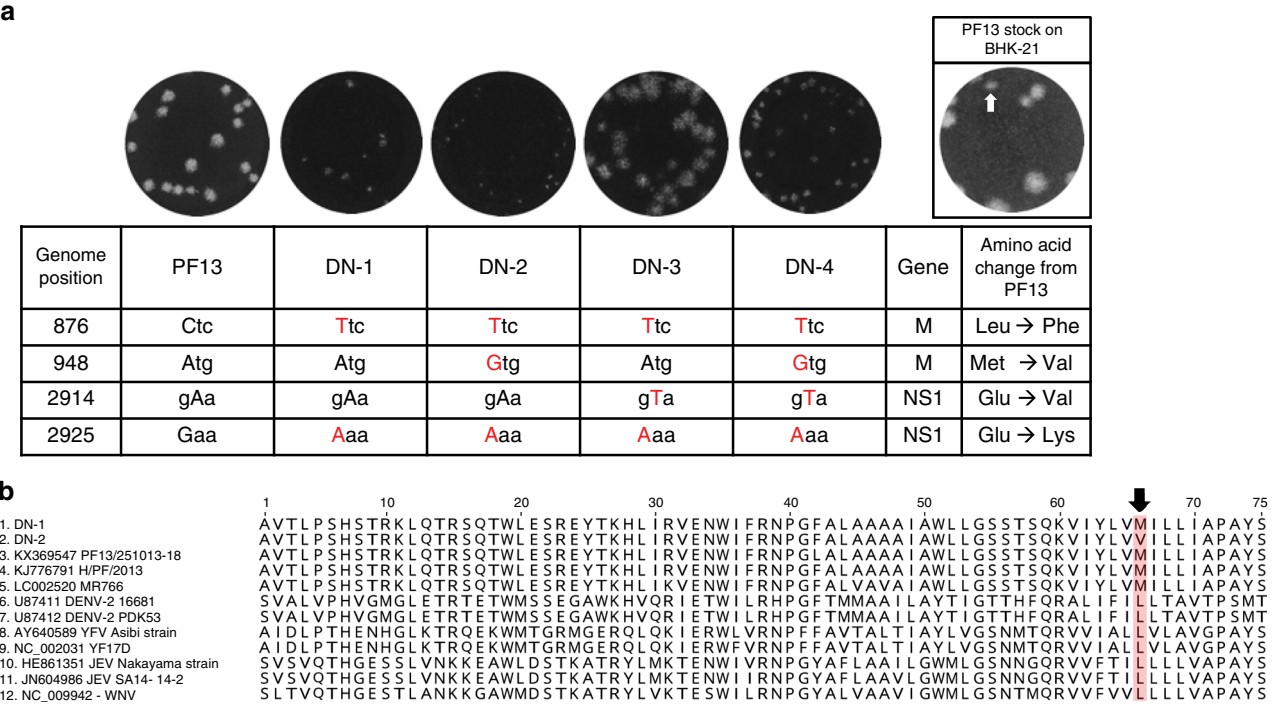

**Fig. 1** Infectious clone-rescued small-plaque ZIKV and unique M protein mutation. **a** Plaque phenotype on Vero cells and mutations of infectious clone-derived ZIKV, based on sequences of small-plaque variants (white arrow) purified from C6/36-derived PF13 stock. **b** The amino acid sequences of various ZIKV strains of different lineages (Asian and African) and both wild-type and vaccine strains, if available, of various mosquito-borne flaviviruses were aligned and the unique M66V mutation in DN-2 is highlighted in red and indicated with an arrow

**Table 1 Consensus changes in DN-2 and DN-1 after serial passage in Vero cells**

| Virus | Position | Passage number | | | | | | | Gene | Mutation |
|---|---|---|---|---|---|---|---|---|---|---|
| | | 1 | 2 | 3 | 4 | 5 | 6 | 7 | | |
| DN-2 | 948 | G | G | G | G | G | G | G | *M* | — |
| | 2904 | A | A | A | T | T | T | T | *NS1* | T → S |
| DN-1 | 2449 | T | T | T | C | C | C | C | *E* | L → S |
| | 2925[a] | A | A | A | A | G | G | G | *NS1* | K → E |
| | 4197 | C | C | C | T | T | T | T | *NS2A* | Silent |
| | 8772 | G | G | G | A | A | A | A | *NS5* | E → K |

[a]Reversion to PF13 sequence

growing strain that does not elicit an IFN response such as the disease-causing DENV3 LAV candidate PGMK30 could still form small plaques[22,23]. To determine if the small-plaque size of DN-2 was due to the IFN-restricted propagation on a cell monolayer, we carried out plaque assay using IRF-3 silenced BHK-21 cells. Similar to our previous findings with PDK53[22], IRF3 silencing (Fig. 3, Supplementary Fig. 2) resulted in increased plaque sizes of DN-2. No significant difference, however, was observed for either DN-1 or the wild-type ZIKV H/PF/2013, a pathogenic strain used in multiple animal models[24] (Fig. 3). This finding indicates that IRF3, an important signaling molecule in IFN induction[25], restricted infection and spread of DN-2 thereby causing its small-plaque phenotype.

**Effective moDCs infection with innate immune activation**. To induce robust B- and T-cell responses, DN-2 would need to infect dendritic cells (DCs) effectively as activation of DCs is a critical event to initiate migration from the site of vaccine inoculation to draining lymph nodes where MHC-mediated presentation of cytoplasmic-synthesized peptide antigens to T cells occurs[26–28]. We found that DN-2 was able to infect monocyte-derived DCs (moDCs) and produce infective virions more effectively than DN-1 (Fig. 4a, b). Furthermore, microarray analysis of infected moDCs demonstrated that DN-2 was also able to induce a similar set of innate immune response genes as YF17D virus (Fig. 4c, d). This is interesting as induction of innate immune response was shown to be a correlate of YF17D immunogenicity in a human trial, although this clinical observation was made through measuring gene expression on total RNA extracted from whole blood instead of DCs[12].

In contrast, DN-2 replicated to a significantly lower level than either DN-1 or wild-type ZIKV during infection in monocyte-derived macrophages (mDMs) (Supplementary Fig. 3a, b). We are uncertain about the significance of this observed difference in infection outcomes in moDCs and macrophages at this time. However, severe dengue is thought to result from increased infection of macrophages in lymphoid organs[29]. Consequently, the different infection outcome could prove to be a useful property for LAV as it suggests that DC-mediated antigen presentation can occur effectively without further virus amplification in lymph node-resident macrophages.

**Infectivity and immune response in endothelial cells**. As infection of ECs is a necessary first step for the virus to cross from systemic circulation into vital organs, such as the central nervous system (CNS), we next examined DN-2 infection in ECs. Our results showed that DN-2 produced significantly lower intracellular viral RNA and plaque titers compared to either DN-1 or the wild-type ZIKVs in human umbilical vein ECs (HUVECs) (Supplementary Fig. 4a, b), which are mature ECs shown to be susceptible to ZIKV infection[30].

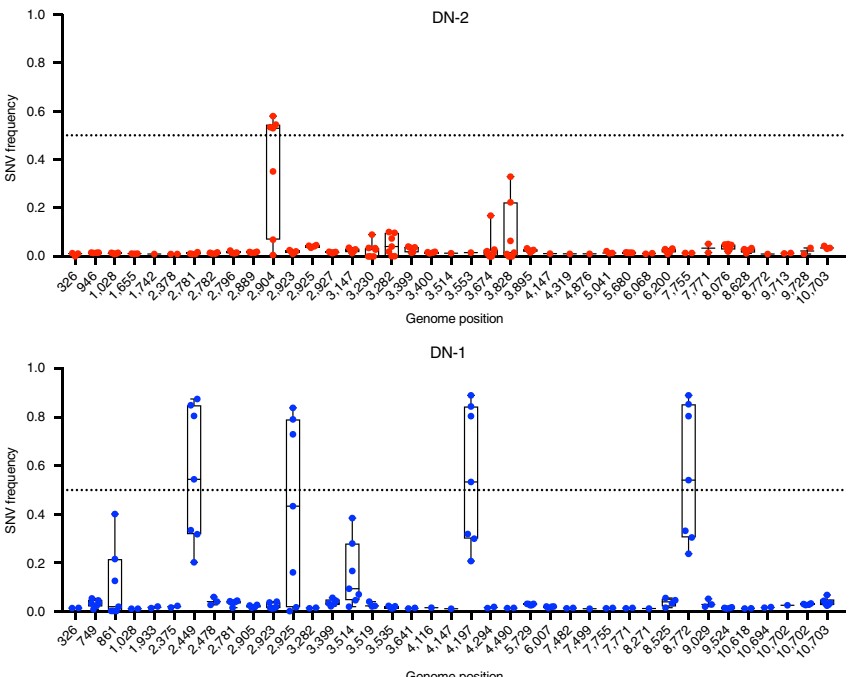

**Fig. 2** Genome stability of DN-1 and DN-2 after serial passage. Whole-genome sequencing of DN-1 and DN-2 at each of seven serial passages was performed using next-generation sequencing to determine stability of the M66V mutation and identify additional mutations, if any, after each serial passage. Single-nucleotide variants (SNVs), with >0.01 (1%) frequency, at different genome positions at each serial passage are plotted to determine genome stability. Each data point represents the frequency at a serial passage. Box and whiskers plots show the range of frequencies of SNVs detected over the serial passages, with the box representing the 25th–75th percentiles while the whiskers represent the maximum and minimum frequencies detected. Dotted line at the SNV frequency of 0.5 represents the point above which a variant is recognized as the consensus sequence

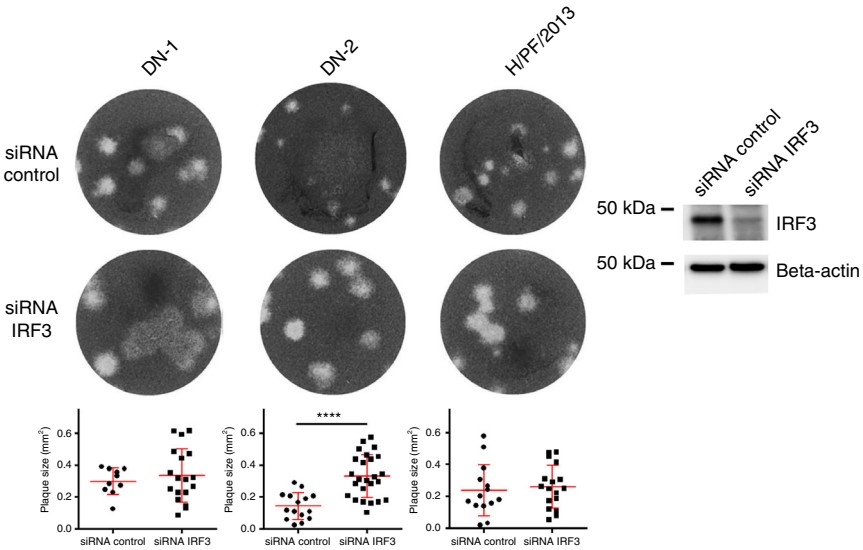

**Fig. 3** Immune-restriction of DN-2 replication confers small-plaque phenotype. Plaque sizes of DN-1, DN-2, and H/PF/2013 after siRNA knockdown of IRF3 in BHK-21 cells. Western blot analysis of cells was performed 48 h after siRNA transfection to determine knockdown efficiency as shown in the cropped blots (representative data of three experiments). Error bars represent s.d. ****$p < 0.0001$ in unpaired $t$ test

We next tested the infection of similar ZIKV strains in human embryonic stem cell-derived endothelial progenitor cells (hESC-derived EPCs) that were differentiated on laminin 521-coated plates in a chemically defined, xeno-free medium[31]. Compared to HUVECs, these EPCs have previously been shown to be functional with high expression levels of markers that are specific for early EC lineage (e.g., CD34, VEGFR2, CD31, VE-cadherin)[31].

Despite this difference in maturity, DN-2 consistently showed reduced levels of infection in the EPCs as compared to DN-1 and wild-type ZIKV strains (Fig. 5a, b).

Furthermore, infection in EPCs also resulted in robust induction of cytokines such as *CXCL10*, type I IFN as well as IFN-stimulated genes (ISGs), such as *IFIT2*. In contrast, DN-1 and wild-type strains H/PF/2013 and Paraiba, both of which have

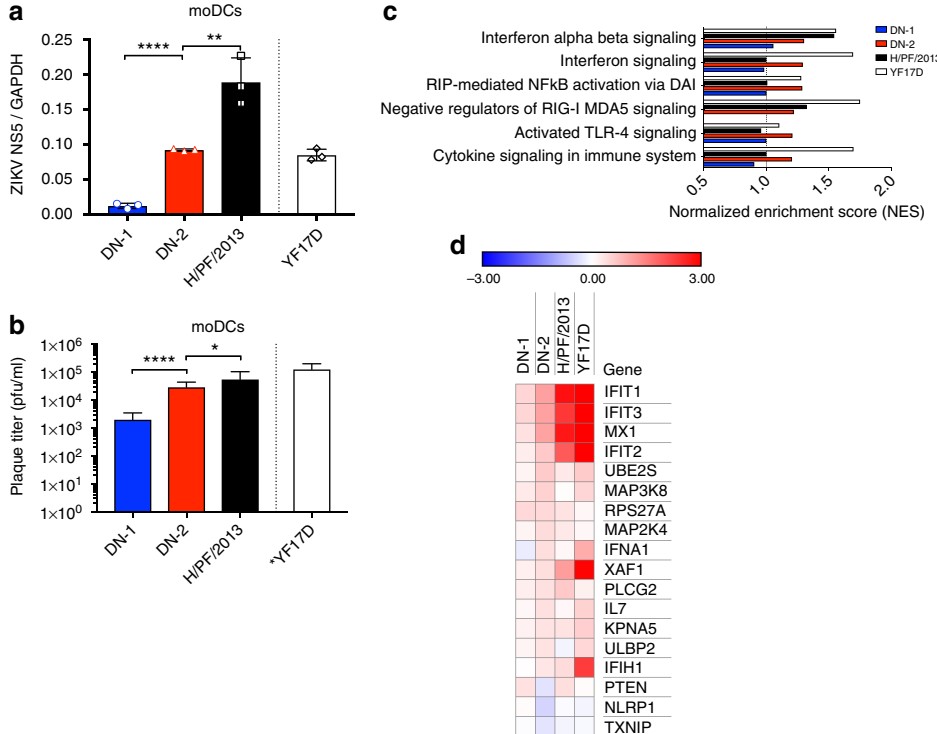

**Fig. 4** Efficient replication and immune gene activation in monocyte-derived dendritic cells (moDCs). Viral replication in moDCs at 24 h post infection as determined by **a** qPCR ($n = 3$, representative plot of two experiments) and **b** plaque assay ($n = 6$) on Vero (results of two experiments combined, YF17D plaque assay was performed on BHK-21). **c, d** Microarray was performed on infected moDCs ($n = 3$) after 24 h post infection and analysis was done in comparison to uninfected cells ($n = 3$). **c** GSEA analysis of the most enriched signaling pathways in DN-2 compared to DN-1, H/PF/2013, and YF17D after 24 h post infection in moDCs. Normalized enrichment scores (NES) were computed by GSEA ($n = 3$). **d** Heat map of differentially expressed immune-related genes. Values represented as $\log_2$(fold change) compared to uninfected cells. Error bars represent s.d. *$p < 0.05$, **$p < 0.01$, ***$p < 0.001$, ****$p < 0.0001$ in unpaired $t$ test. n.s. not significant

been shown to cause organ and vertical transmission in animal models[24], infected EPCs at higher rates, but they did not induce a significant level of IFN or ISG responses in EPCs (Fig. 5c, Supplementary Fig. 4c, d).

Type III IFNs have been shown to protect against CNS infections by West Nile and YF viruses through exerting its antiviral effects on the ECs at the blood-brain barrier[32,33]. More recently, type III IFNs have also been shown to protect against ZIKV infection of the placenta and vertical transmission[34–37]. In our in vitro system, DN-2 induced a higher expression of type III IFNs (IL28A, IL28A/B) than DN-1, H/PF/2013 or Paraiba strains, although these differences were not statistically significant ($p > 0.05$, Student's $t$ test), (Supplementary Fig. 4c). However, DN-2 showed increased susceptibility to the antiviral effects of IFN-λ1 compared to either H/PF/2013 or DN-1 (Fig. 5d) that compensates for the lack of major difference in type III IFN expression. Collectively, our data suggests that the refractoriness of DN-2 to infect ECs, due at least in part to the inhibitory effects of types I and III IFNs, could make DN-2 attenuated in causing persistent organ infections.

**Attenuation in immunocompetent human cells and *Aedes aegypti*.** Besides moDCs, mDMs, HUVECs, and EPCs, we also compared the infectivity of DN-2 against the other ZIKV strains in other immunocompetent human cells. The intracellular viral load in the immunocompetent liver cell line HuH-7 after 24 h of infection was significantly lower in DN-2 compared to either DN-1 or H/PF/2013 (Fig. 6a). Interestingly, DN-2 also showed the lowest infection rate in a fetal-derived, diploid human lung fibroblast cell strain, MRC-5 (Fig. 6b).

Furthermore, DN-2 also caused decreased infection and replication rates in *A. aegypti* mosquitoes. Mosquitoes fed on blood spiked with DN-2 resulted in 5 $\log_{10}$ lower levels of viral RNA compared to those fed with a similar amount of the highly infective and transmissible Paraiba strain[38] (Supplementary Fig. 5). The detectable DN-2 infection in *A. aegypti* will need further vector competence investigation although it is unlikely to be epidemiologically significant. This is because the viremia level of DN-2 is significantly lower than wild-type ZIKV (as shown below). A mosquito feeding on a vaccinated individual would thus ingest fewer DN-2 than wild-type ZIKV that, coupled with the reduced vector infection rates, minimizes the likelihood of vaccine transmission[38,39].

**Safety and immunogenicity in the A129 mouse model.** To test if these in vitro properties would translate to attenuated in vivo infection, we infected IFN-α/β receptor-deficient A129 mice with the various strains of ZIKV[24]. This mouse model recapitulates some of the disease and tissue tropism features of ZIKV and has been used to assess other ZIKV vaccine constructs[40–42]. Male A129 mice were intraperitoneally (i.p.) inoculated with $10^3$ and $10^4$ p.f.u. of DN-1, DN-2, or H/PF/2013. H/PF/2013 infection was uniformly lethal at both doses, causing rapid weight loss and paralysis in infected mice. In contrast, mice that received either DN-1 or DN-2 survived infection at both doses without neurological symptoms. However, DN-1 but not DN-2-infected animals showed transient weight loss between days 5 and 10 at $10^4$ p.f.u. dose (Fig. 7a, b).

Consistent with the survival data, H/PF/2013 produced the highest peak viremia levels at around $10^9$ copies per ml for both

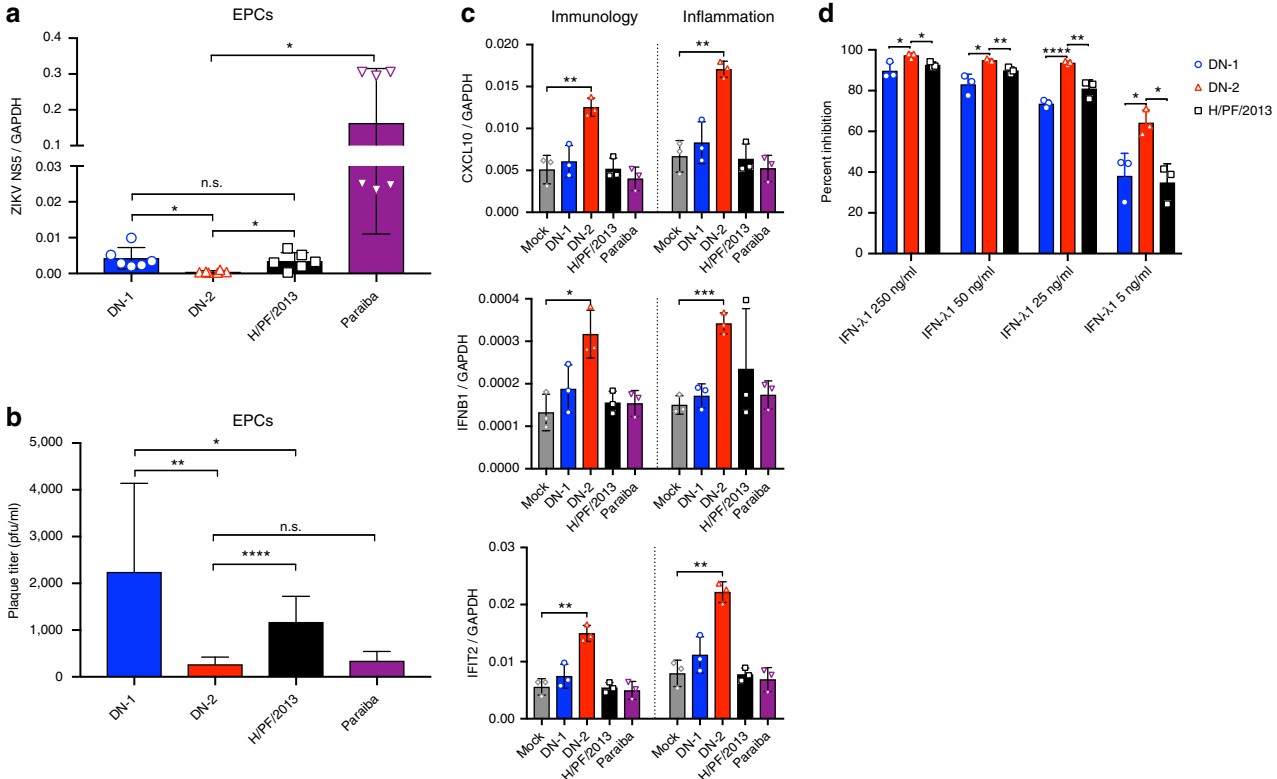

**Fig. 5** Attenuated infectivity on endothelial progenitor cells is accompanied by innate immune activation and interferon sensitivity. Viral replication of different ZIKV strains in human embryonic stem cell-derived endothelial progenitor cells (hESC-derived EPCs) was determined by **a** qPCR ($n = 6$) and **b** plaque assay ($n = 9$) (results are combined from three experiments). **c** NanoString nCounter analysis was performed on infected EPCs at 24 h post infection ($n = 3$) using both the Immunology and Inflammation prebuilt panels. Counts obtained for CXCL10, IFNB1, and IFIT2 were normalized to GAPDH counts. **d** Percent inhibition of virus replication as determined by viral RNA qPCR on Vero culture supernatant after 48 h of infection with co-treatment at different concentrations of IFN-λ1 ($n = 3$ per data point, representative data of two experiments). See also Supplementary Fig. 3. Error bars represent s.d. *$p < 0.05$, **$p < 0.01$, ***$p < 0.001$, ****$p < 0.0001$ in unpaired $t$ test. n.s. not significant

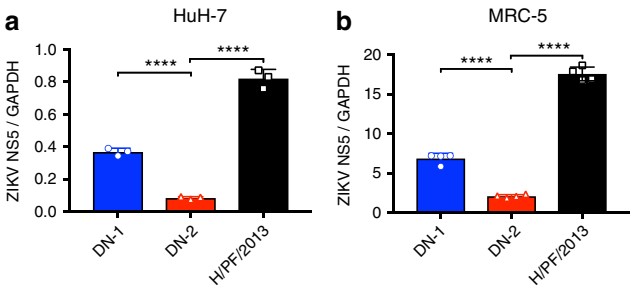

**Fig. 6** Intracellular viral replication determined by qPCR after 20 and 24 h post infection in **a** HuH-7 cell line ($n = 3$) and **b** MRC-5 cell strain ($n = 4$). Error bars represent s.d. ****$p < 0.0001$ in unpaired $t$ test

doses of inoculum. Viremia levels remained detectable at death in animals infected with H/PF/2013 (Fig. 7c, d). Peak DN-1 viremia was similar to H/PF/2013 at $10^4$ p.f.u. of inoculum but 1.5 $\log_{10}$ lower with $10^3$ p.f.u. In contrast, peak DN-2 viremia was the lowest at approximately $10^7$ copies per ml at both doses and waned to below detection limit 8 days after inoculation (Fig. 7c, d). These findings demonstrate that the in vivo replication of DN-2 is attenuated relative to either DN-1 or wild-type ZIKV.

The neutralizing antibody titers at day 21 post inoculation trended with the viremia profiles of DN-1 and -2, where the lower DN-2 viremia resulted in a 0.5 $\log_{10}$ lower neutralizing antibody titer than DN-1 infected animals (Fig. 7e). This finding is similar to our clinical findings with YF17D vaccination where longer viremia was associated with higher neutralizing antibody titer[16].

However, the 2.5 $\log_{10}$ PRNT$_{50}$ titers produced from DN-2 infection proved adequate to fully protect the animals from a $10^4$ p.f.u. H/PF/2013 challenge. All mice pre-inoculated with either DN-1 or -2 survived this lethal challenge with no weight loss. In contrast, rapid weight loss with 60% mortality was observed in the control animals (Fig. 7f, g). Importantly, quantitative PCR (qPCR) was not able to detect viremia in animals pre-inoculated with DN-1 and DN-2, whereas control animals showed a viremia profile similar to that of H/PF/2013 inoculation in younger naive mice (Fig. 7h).

These show that the replication of DN-2 is sufficiently attenuated despite the lack of type I IFN signaling in the A129 mice model and was able to elicit a protective response against a lethal challenge.

**Decreased organ dissemination in A129 mouse model.** We next tested if pre-inoculation with DN-2 could also protect against viral infection in organs that, in human infections, either result in disease (brain) or sexual transmission of the virus (testes and kidney)[17]. Although ZIKV is known to also infect other organs[17], such as the eye, the clinical significance of persistent infection in these organs is unclear. To determine the rates of infection in these key organs following H/PF/2013 challenge, we collected these organs at day 15 post challenge or when control mice had to be killed. Viral infection in these organs were tested by qPCR followed by sequencing of the *prM* gene to distinguish DN-1 and DN-2 from the challenge virus. In control animals, H/PF/2013 ZIKV RNA was detected in all tested organs while DN-1 and DN-

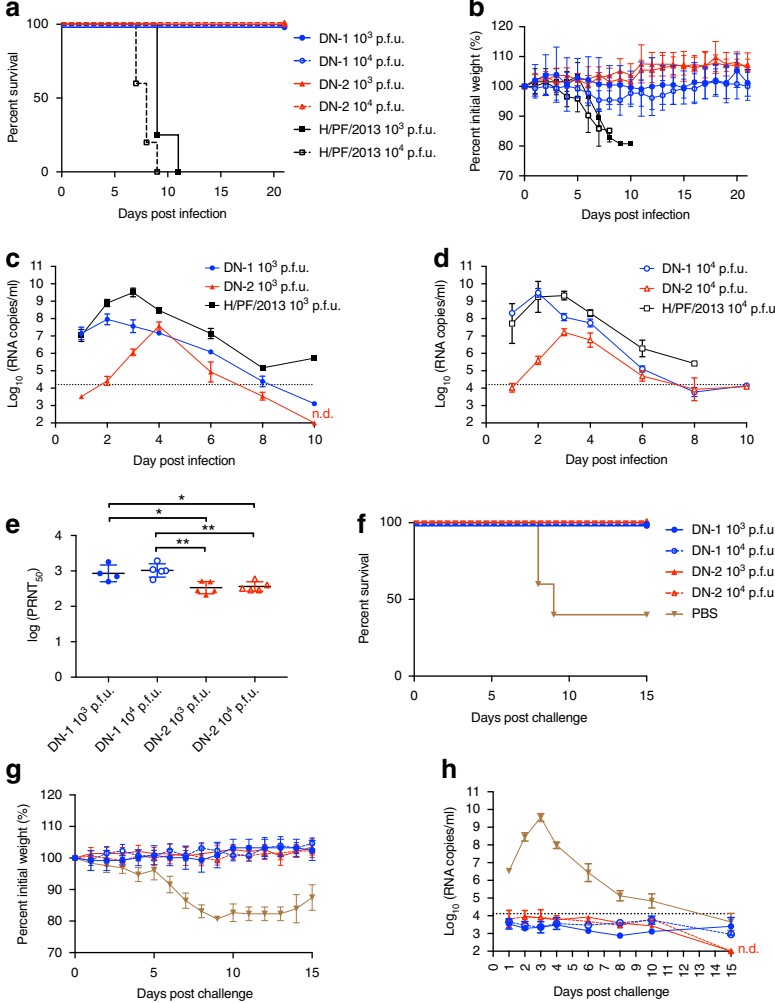

**Fig. 7** DN-2 displays attenuated replication in A129 mouse model and elicits protective responses against lethal wild-type ZIKV challenge. Male A129 mice were inoculated i.p. with $10^3$ or $10^4$ p.f.u. of DN-1, DN-2, or H/PF/2013 ($n = 5$ per group except $n = 4$ for mice that received $10^3$ p.f.u. of DN-1 and H/PF/ 2013) and monitored for **a** survival, **b** weight changes and viremia levels after infection with **c** $10^3$ and **d** $10^4$ p.f.u. of each virus. **e** Immune sera from surviving mice obtained at 21 days post inoculation to determine neutralizing antibody titers using PRNT and the log 50% neutralizing titers [log(PRNT$_{50}$)] are as shown. These mice, together with a group of mice that received PBS at day 0 ($n = 5$), were challenged with $10^4$ p.f.u. of H/PF/2013 i.p. at 21 days post inoculation. **f** Survival, **g** weight changes, and **h** viremia levels were tracked for 15 days or until euthanasia for control mice with >20% weight loss. Error bars represent s.d. *$p < 0.05$, **$p < 0.01$ in unpaired $t$ test. Dotted lines on viral quantification graphs represent limit of detection. n.s. not significant, n. d. not detectable

2 pre-inoculation prevented accumulation of viral RNA within the spleen (Fig. 8a). However, DN-1 RNA was detected in the testes and brains of all DN-1-inoculated mice, albeit at lower levels in the brain compared to controls (Fig. 8a–c). Comparatively, only 1 (20%) and 2 (40%) out of 5 mice that received $10^3$ and $10^4$ p.f.u. of DN-2, respectively, had detectable DN-2 RNA in the testes (Fig. 8a–c). Despite the positivity, the viral load in these organs were significantly lower than in control animals or those inoculated with DN-1 (Fig. 8b). Furthermore, DN-2 RNA was negative in all other organs tested (Fig. 8a–c). Taken together, these findings suggest that pre-inoculation with DN-2 fully prevented H/PF/2013 viremia and organ infection, with only low-level persistent infection in the testes in a small proportion of animals. This might not be too surprising given the importance of the type I IFN response in controlling DN-2 in both the hematopoietic and EC compartments that is absent in these type I IFN receptor-deficient mice.

To further investigate the extent of organ dissemination in an earlier stage of infection, 3 male mice were infected with $10^4$ p.f.u.

of each virus i.p. and organ viral load measured 8 days later. To obtain an indication of viral enrichment in the organs, we normalized the viral RNA levels in the organs (Supplementary Fig. 6a) against the viremia levels (Supplementary Fig. 6b) at the time of killing. As expected, the blood-rich spleen showed no organ enrichment relative to viremia for all viruses. H/PF/ 2013 showed the highest enrichment of virus in the brain (Fig. 8d). Two of the three DN-1 infected animals showed a trend toward viral enrichment in the brain although the mean levels were not significantly different compared to DN-2-infected animals. In the testes, both H/PF/2013 and DN-1 RNA were similarly enriched whereas DN-2 RNA levels were near parity (Fig. 8d).

While the challenge study demonstrated the ability of DN-1 and DN-2 to elicit protective immunity, the higher infectivity of DN-1 on ECs appears to indicate a greater propensity for invasion and persistence in the brains and testes of A129 mice than DN-2 from an early time point. This lends proof to the concept that reduced infectivity on ECs could translate to decreased organ dissemination of ZIKV.

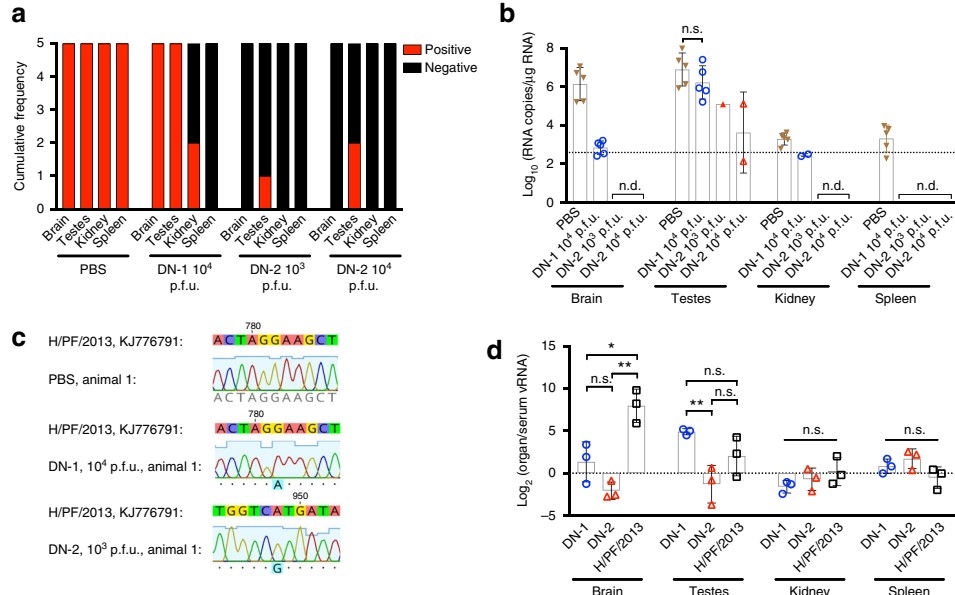

**Fig. 8** DN-2 demonstrates decreased organ infection. **a** Frequency of viral RNA detection and **b** levels of viral RNA detected in the brains, testes, kidneys, and spleens using at 15 days post challenge or at killing for animal study in Fig. 5. **c** Sequence of viral RNA detected in testes samples were determined using Sanger sequencing to distinguish identity of virus. **d** Male mice ($n = 3$ per group) were infected with $10^4$ p.f.u. of DN-1, DN-2 and H/PF/2013 i.p., killed at 8 days post infection and viral loads in the brains, testes, kidneys, and spleens were quantified using qPCR with normalization against viremia. Error bars represent s.d. *$p < 0.05$, **$p < 0.01$ in unpaired $t$ test. Dotted lines on viral quantification graphs represent limit of detection or when $y$-value = 0 in normalized values. n.s. not significant, n.d. not detectable

**Attenuated maternal–fetal transmission in A129 dams**. In the placenta, fetal ECs and the trophoblasts form the barrier between maternal and fetal circulation[43], all of which have been shown to be highly susceptible to both African and Asian lineage ZIKV infection[30,34,36,44–46]. Since DN-2 has decreased tropism for ECs (Fig. 5, Supplementary Fig. 4) and cytotrophoblasts differentiate toward an endothelial phenotype[47], we posit that DN-2 would also be attenuated in its ability to cross the placental barrier from maternal circulation to infect the fetuses.

Female A129 mice were infected with $10^3$ p.f.u. of DN-1, DN-2 or H/PF/2013 intravenously (i.v.) 6 days after a 4-day mating period and killed at 7 days post infection (Fig. 9a). To assess the maternal viral burden, maternal viremia was measured on days 2 and 7 post infection. Significantly lower levels of viremia were detected in animals infected with DN-2 compared to either DN-1 or H/PF/2013 (Supplementary Fig. 7a).

We next evaluated the placental and fetal infection in these mice. A single mouse from each group did not show presence of conceptus and was excluded from analysis. Unsurprisingly, fetuses of pregnant mice infected with H/PF/2013 succumbed to infection; only one animal showed surviving fetuses out of three pregnant dams. A single dead fetus was found in one of the four dams infected with either DN-1 or DN-2. Fetuses of DN-1 and DN-2-infected dams did not show gross pathology and fetal absorption rates were similar to uninfected dams, as compared to high rates of fetal death when dams were infected with H/PF/2013 (Fig. 9b, c). No difference in weight distribution of surviving fetuses was observed in either DN-1- or DN-2-infected dams compared to uninfected controls; the heterogeneity in fetal weight distribution within each group of animals is likely a reflection of the different stages of fetal development due to non-synchronized mating (Fig. 9d). The amounts of virus RNA detected in the placenta trended with fetus size (Supplementary Fig. 7b). Indeed, the placenta in H/PF/2013-infected dams may be smaller relative to those from either DN-1 or -2-infected animals due to intra-uterine deaths. Importantly, although ZIKV RNA could be

detected in the placenta, viral RNA levels in the fetuses were below the limit of detection in fetuses from DN-1 and DN-2-infected dams except for one fetus in each group; the viral load of DN-1 was higher than DN-2 in those animals (Fig. 9e).

Collectively, these findings suggest that a ZIKV strain with lowered viremia and attenuated in EC infection could reduce the risk of trans-EC spread into critical organs and also maternal–fetal transmission.

## Discussion

LAV development is a time-consuming process although there is now increasing knowledge on attenuating mutations that could enable an engineered approach for accelerated LAV development. Attenuation mutations in different flaviviruses have been identified[48] and applied, as exemplified by the recently published 3′ UTR deletion ZIKV LAV candidates that have shown useful efficacy and safety in both murine and primate models[41,42]. However, such information is still limited and a mutation that attenuates one flavivirus may not necessarily attenuate another flavivirus. A systematic approach to screen for attenuated viruses thus still has an important place in vaccine development as it enables the discovery of novel attenuating mutations for the isolation of LAV with unique and beneficial properties.

ZIKV, like DENV and YFV, are flaviviruses spread by *Aedes* mosquitoes. Infection with any of these viruses result in transient viremia that in a proportion of individuals produce acute febrile illness[49]. Although the severe infection outcome differs between these viruses, the immune responses needed to protect against infection may be similar, given the close genetic relationship of these viruses and the similar mode of transmission. Lessons learnt from DENV and YFV vaccine development should thus be used to guide the choice of ZIKV vaccine to avoid pitfalls of the current dengue vaccine. Indeed, vaccine viremia may be an important determinant of flaviviral immunogenicity. Detectable viremia was highest for the DENV4 component of the chimeric yellow

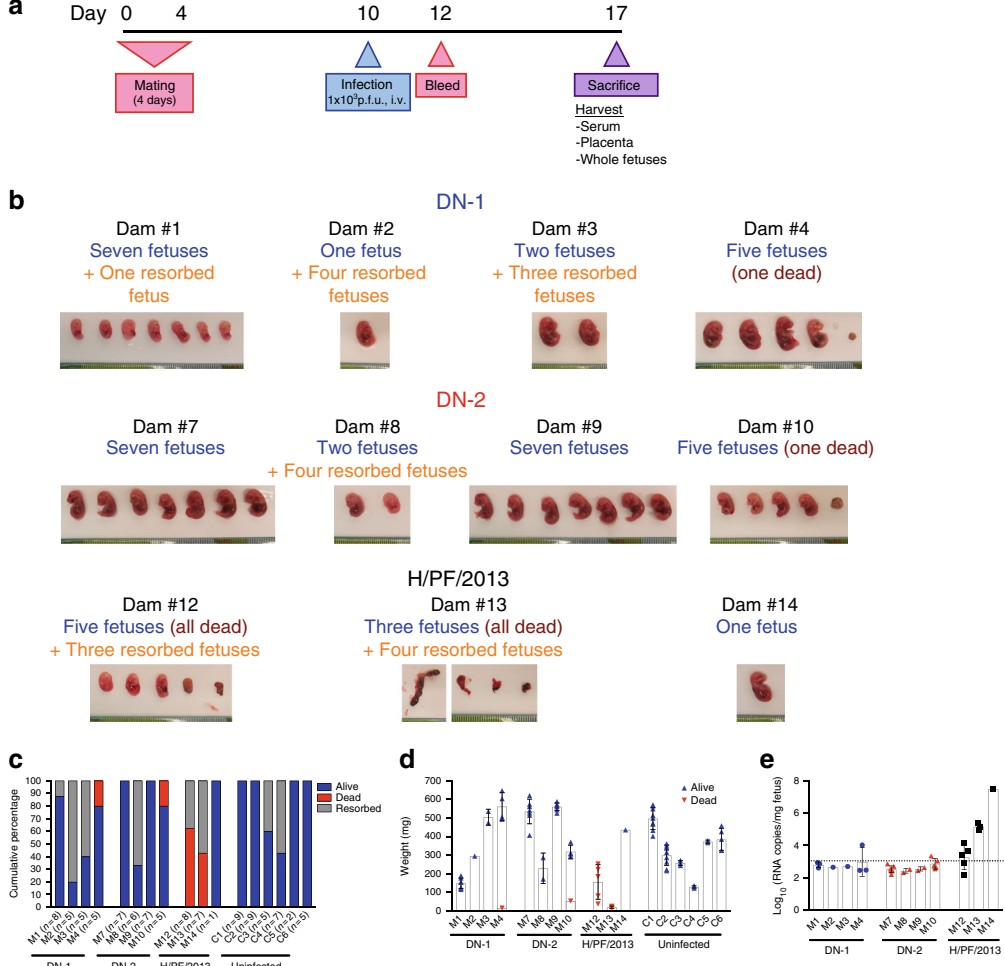

**Fig. 9** Decreased vertical transmission during DN-2 infection of pregnant A129 mice. **a** DN-1, DN-2 and H/PF/2013 were tested in a pregnant mouse model according to the timeline shown. **b–e** The dams ($n = 5$ for DN-1 and DN-2, $n = 4$ for H/PF/2013) were killed at day 7 post infection. Of these mice (infected dams numbered M1 to M14 with $n =$ number of fetuses indicated, uninfected controls numbered C1–C6), four each in the DN-1 and DN-2-infected dams and three in the H/PF/13-infected group were pregnant and the placenta and fetuses were also collected. **b** Fetuses of infected pregnant dams showing absence or presence of gross pathological changes caused by vertical ZIKV transmission. **c** Survival and **d** weight of fetuses were determined and **e** viral RNA quantification was performed on whole fetus homogenates using qPCR. See also Supplementary Fig. 6. Error bars represent s.d. Dotted lines on viral quantification graphs represent limit of detection

fever dengue tetravalent vaccine, followed by the DENV3 and DENV1 components, whereas the DENV2 component of the vaccine produced no detectable viremia[50]. The proportion of detectable viremia thus trended distinctly with the observed serotype-specific efficacy of this vaccine[51]. Furthermore, both the inactivated Japanese encephalitis vaccine and the tick-borne encephalitis vaccine, which are also flaviviruses but transmitted via non-*Aedes* vectors, do not produce long-lasting immunity despite multiple dosing as regular boosters are recommended by the WHO[52,53]. These findings contrast with the single-dose recommendation by WHO for YF LAV[10,54], vaccination with which produces detectable viremia in 100% of individuals[16].

The findings of our study highlight the possibility to develop a ZIKV LAV candidate that bears the virological and immunological traits of YF17D and PDK53 but with the additional safety feature of reduced persistent organ infection. We derived a small-plaque variant, DN-2 that also exhibits reduced infection rates of critical organs involved in severe disease, human-to-human transmission, and possibly even vertical transmission. Critically, the absence of neurological symptoms or detection of virus in the brain after DN-2 inoculation despite recent findings of cytotoxic T-cell-mediated neuropathology consequent of neuroinvasion in

*Ifnar*$^{-/-}$ mice[55] suggests sufficient attenuation of DN-2. Despite the reduced organ infection, the levels and duration of DN-2 viremia was sufficient to elicit protective immunity against a lethal wild-type ZIKV challenge.

Despite significant reduction in organ dissemination, DN-2 still caused persistent testicular infection in a small proportion of A129 mice. This outcome may constitute some level of safety concern. However, given the importance of the type I IFN response triggered by DN-2 in both hematopoietic and ECs (Figs. 4 and 5), which also controlled the spread of this virus in a cell monolayer (Fig. 3), the persistent testicular infection may be due directly to deficient type I IFN response. Indeed, type I IFN response has been found to be important in controlling the measles, mumps, and rubella vaccine infection as children with primary immunodeficiencies in type I IFN signaling succumbed to these LAVs[56,57]. That the infection outcome with DN-2 infection in A129 mice is attenuated would further suggest that this ZIKV strain is sensitive to the antiviral effects of types II and III IFN. The latter is important as IFN-λ has been shown to control vertical transmission of ZIKV and neuroinvasion by other neurotropic flaviviruses[32–37,45]. Despite an intact type III IFN in A129 mice, however, the prevention of viral dissemination and

persistent infection may require the action of all three types of IFN as shown by a recent report where IFN-λ response prevented persistent norovirus infection but type I IFN is nonetheless needed to prevent systemic viral dissemination[58]. Notwithstanding this line of thinking, however, detailed safety investigations in larger pre-clinical models will need to be studied to move any ZIKV LAV along the clinical development pathway.

DN-2 differs from DN-1 by a single amino acid substitution in the M protein. How such a mutation, which is in the trans-membrane region of the M protein (with reference to the DENV M protein[59]), attenuated ZIKV remains to be defined. The importance of the flavivirus prM protein in affecting the host cell permissibility is only beginning to be explored with a recent demonstration of a S139N substitution in the prM protein as a likely contributor of increased microcephaly occurrences in the ZIKV outbreaks in French Polynesia and the Americas[60]. A more recent study also described a R15Q amino acid change in the M protein that decreased plaque sizes and slightly decreased virulence in IFNAR1-knockout C57BL/6 mice[61]. The M protein M66V single mutation we describe here is stable up to seven passages and also stabilized the viral genome considerably as compared to the multiple variations found in DN-1 upon serial passaging in Vero cells. Interestingly, this mutation also produced significant differences in infectivity in cell lines/strains and primary cells such as moDCs, macrophages, ECs, and even fetal fibroblasts. It also resulted in increased induction of IFN expression in infected cells and increased sensitivity to the anti-viral effects of IFN-λ1. This IFN susceptibility could underlie the attenuated ability of DN-2 to cross EC barriers for organ infection, sexual and even vertical transmission, which could be an important safety feature of any ZIKV LAV.

Our findings underscore the outcome of another ZIKV LAV development. In developing 3′UTR-deletion LAV candidates, the authors first concluded that the Δ10 mutant was the best candidate as it produced the lowest viremia levels in A129 mice[41]. However, the low level or even absence of viremia in Rhesus macaques resulted in low neutralizing antibody titers, which were significantly boosted after wild-type ZIKV challenge, suggesting a lack of complete protection against wild-type ZIKV infection[42]. In contrast, the more replicative Δ20 mutant elicited higher levels of antibody response and correlated with complete protection from infection[41,42]. With increased replication, the risk of organ dissemination and infection persistence could theoretically be increased. Our findings thus suggest the possibility of engineering a ZIKV LAV that produces sufficient viremia to elicit protective immunity while minimizing organ dissemination through reduced EC infectivity. Further pre-clinical studies of DN-2 safety and immunogenicity in non-human primates are thus warranted.

In conclusion, our study provides a framework to approach the development of LAVs against ZIKV and perhaps even other flavivirus that spread beyond the systemic and lymphatic circulations for either persistent or disease-inducing organ infection.

## Methods

**Cells and virus cultures**. BHK-21 (ATCC CCL-10), C6/36 (ATCC CRL-1660), Vero (ATCC CCL-81), HuH-7 (from Duke Cell Repository), HEK293T (ATCC CRL-3216), and MRC-5 (ATCC CCL-171) cells were purchased from the American Type Culture Collection (ATCC) unless otherwise stated. BHK-21 and C6/36 were cultured in RPMI 1640 (Gibco); Vero, HuH-7, and HEK293T were cultured in DMEM media (Gibco); MRC-5 was cultured in EMEM (ATCC), all of which were supplemented with 9% fetal calf serum (HyClone). All cell lines tested negative for mycoplasma. Wild-type ZIKV strains, namely PF13/251013-18[62] (KX369547, at fourth passage in Vero, gift from Didier Musso at Institut Louis Malardé), H/PF/2013[63] (KJ776791, from European Virus Archive) and Paraiba (KX280026, gift from Pedro F. C. Vasconcelos at Instituto Evandro Chagas) ZIKV strains were passaged on C6/36. YF17D virus was obtained from passage of Stamaril (Sanofi Pasteur) in Vero as previously described[16]. After infection, cells are maintained in maintenance media which contain 2% FBS, 100 U/ml penicillin, and

100 μg/ml streptomycin. Viral titers were determined by plaque assay on BHK-21 and/or Vero. All experiments performed with ZIKV were based on plaque titers obtained on Vero while YF17D was titered on BHK-21.

**Plaque assay and plaque purification**. Plaque assay was performed on BHK-21 or Vero as previously described[64] using maintenance media with RPMI 1640 for BHK-21 and DMEM for Vero. For plaque purification, BHK-21 was seeded in six-well plates 2 days before infection with ~15 p.f.u. of PF13 per well in 100 μl. After 1-h adsorption of inoculum at 37 °C, inoculum was removed and overlaid with maintenance media containing 0.9% agarose (BD BaculoGold Plaque Assay Agarose). Five days later, plaques of various sizes were isolated for Sanger sequencing. Isolated plaques were also passaged in Vero followed by plaque assay on BHK-21 cells for plaque size verification.

**Infectious clone generation and virus recovery**. RNA was extracted from PF13 (four passages in Vero and once in C6/36) using TRIzol LS (Invitrogen) and full-length sequencing was determined using NGS. Complementary DNA (cDNA) synthesis was performed using SuperScript III first-strand synthesis Kit (Invitrogen) as per manufacturer's instructions. Five PCR fragments of around 2000 nucleotides long were generated from cDNA using primer pairs (design based on NGS-derived genome sequence) in Supplementary Table 1 with NEB Q5 Hot-Start high-fidelity 2× Master Mix (New England Biolabs). Fragments were gel-purified with MinElute gel extraction Kit (Qiagen) after agarose gel electrophoresis and TA cloning was performed into pGEM-T Easy Vector (Promega). Plasmids isolated were sequenced using Sanger sequencing. Plasmids used either contain sequences required or have required sequences introduced using QuikChange II site-directed mutagenesis Kit (Agilent). All five viral genome fragments for each ZIKV strain were amplified using NEB Q5 hot-start high-fidelity 2× Master Mix (New England Biolabs) from corresponding plasmid templates using corresponding primer pairs. Vector similar to what was previously described[65] (generous gift from Katell Bidet at Singapore-MIT Alliance for Research and Technology) was also amplified using primer pairs shown in Supplementary Table 1. Amplified fragments were then gel-purified and equimolar amounts (0.1 pmole) of each genome fragment and the amplified vector were assembled using NEBuilder HiFi DNA Assembly Master Mix (New England Biolabs) at 50 °C for 60 min to generate infectious clones. Five microliters of assembled mix (containing infectious clone) was transfected into each well of HEK293T cells in a 24-well tissue culture plate using 3 μl per reaction of Lipofectamine 2000 (ThermoFisher) as per manufacturer's instructions. Media containing infectious clone-derived viruses were collected 48 h post transfection and passaged in Vero cells in T25 tissue culture flasks to recover viruses. Virus titers were determined via plaque assay on Vero cells. Viruses used in all experiments were passaged not more than twice in Vero and only after sequences determined to be unchanged using Sanger sequencing.

**Virus sequencing**. NGS libraries were created using the NEBNext ultra directional RNA Library Prep Kit for Illumina (New England Biolabs) and paired-end sequencing was performed on the Illumina MiSeq (2 × 250 bp) and HiSeq 2000 sequencing systems (Illumina). Reads were de-multiplexed and full viral genomes were assembled as previously described[38]. NGS was performed on the PF13 stock and viral progeny from serial passages of DN-1 and DN-2 in Vero. Sanger sequencing was performed on DN-1 and DN-2 virus stocks to ensure sequence integrity prior to experiments, with sequencing performed on PCR amplification fragments generated using primers in Supplementary Table 1. Both viral genome sequences and flaviviral membrane protein sequences were obtained from sequencing results or downloaded from NCBI and analyzed using Geneious software.

**Plaque size determination after siRNA knockdown of IRF3**. BHK-21 cells were transfected with either control small-interfering RNA (siRNA) or siRNA targeting IRF3 (sense: GGAACAAUGGGAGUUCGAAdTdT and antisense: UUCGAA-CUCCCAUUGUUCCdTdT) (SABio) as previously reported[22] using Lipofectamine RNAiMax reagent (Invitrogen) according to manufacturer's instructions. 48 h post transfection, plaque assay was performed as described above on transfected BHK-21 cells, with DN-1, DN-2, and H/PF/2013 strains of ZIKV. Transfection efficiency was determined by Western blot as previously described[64] using anti-IRF3 (Cell Signaling Technology, #4302S) and anti-β-actin (Cell Signaling Technology, #3700) antibodies, both used at 1:1000 dilution. After staining with crystal violet, the plate was scanned using the ImmunoSpot® Analyzer, (Cellular Technology Ltd.), and smart counting was performed with BioSpot 5.0 software. The plaque-counting parameters were adjusted to optimized settings depending on plaque morphology, and plaques on the edges of each well were excluded from the analyses.

**Viral replication in HuH-7 and MRC-5**. HuH-7 cells were seeded at $1 \times 10^5$ cells per well in 24-well tissue culture plates 1 day prior to infection. Cells were infected at multiplicity of infection (MOI) of 1 by adsorbing 100 μl of inoculum on the cell monolayer for 1 h at 37 °C with rocking at 15-min intervals prior to removal of inoculum and replacement of DMEM maintenance media. MRC-5 was seeded at $2 \times 10^4$ cells per well in 96-well tissue culture plates 1 day prior to infection. Cells were infected at MOI of 1 in a 100 μl volume for 6 h at 37 °C. Inoculum was then

removed and EMEM maintenance media was then added. HuH-7 was collected at 20 h post infection while MRC-5 was harvested at 24 h post infection. Cells were washed with PBS before addition of RLT buffer from RNeasy Mini Kit (Qiagen) and frozen at −80 °C until RNA extraction.

**Monocyte-derived dendritic cells and macrophages.** Peripheral blood mononuclear cells (PBMCs) were isolated from venous blood collected via venipuncture performed on a flavivirus-naïve healthy donor with approval from the National University of Singapore's Institutional Review Board (reference number B-15-227) as previously described[64]. CD14$^+$ monocytes were isolated from PBMCs using CD14 microbeads (Miltenyi Biotec) according to manufacturer's protocol. To differentiate these cells into DCs (moDCs), cells were cultured in six-well tissue culture plates in monocyte growth media (RPMI-1640 supplemented with 10% FBS, 100 U/ml penicillin, and 100 μg/ml streptomycin) containing 100 ng/ml of IL-4 (eBioScience), and 50 ng/ml of granulocyte macrophage–colony stimulating factor (GM-CSF) (eBioScience) for 6 days, with refreshing of cytokine-containing media on the third day. moDC were defined as CD14$^-$ (BD, #561116, 1:50), CD86$^+$ (BD, #555658, 1:50), HLA-DR$^+$ (BD, #552764, 1:50), and DC-SIGN$^+$ (BD, #551265, 1:50), using flow cytometry. To obtain mDMs, CD14$^+$ monocytes were cultured in 24-well tissue culture plates in monocyte growth media supplemented with 100 ng/ml of GM-CSF for 6 days, with replacement of fresh cytokine-containing media on the third day. moDCs and mDMs were infected with ZIKV at MOI 1. Inoculum was replaced with monocyte growth media at 6 h post infection. At 24 h post infection, viral progeny present in the media was collected and frozen at −80 °C until plaque assay on Vero cells. Infected cells were washed once in PBS before lysis in RLT buffer from RNeasy Mini Kit (Qiagen) and frozen at −80 °C until RNA extraction.

**Microarray analysis.** moDCs were infected at MOI 1 with DN-1, DN-2, H/PF/2013, or YF17D in triplicates. Ten wells were pooled for each replicate and three replicates were used for each virus strain in the microarray. RNA was extracted using RNeasy Mini Kit (Qiagen). Microarray was performed at the Duke-NUS Genome Biology Facility and cRNA was hybridized to GeneChip Human Gene 2.0 ST Array (Affymetrix). Data normalization was performed using Partek software and quantile-normalized prior to analysis with Gene Set Enrichment Analysis[66] to identify the significantly enriched pathways mapped against the reactome database. A nominal p value of 0.05 was used as the cutoff. Heat maps were plotted using the Morpheus web program developed by the Broad Institute with the log$_2$ values of fold changes.

**Endothelial cells.** HUVECs were freshly isolated from human umbilical cords obtained from KK Women's and Children's Hospital, Singapore, with approval from the Singapore Health Services Centralized Institution Review Board (CIRB ref: 2014/323/D), and EPCs were differentiated from hESCs (WiCell Research Institute) as previously described[31]. HUVECs used in experiments do not exceed four passages. Both types of ECs were plated on laminin-521-coated 24-well tissue culture plates (LN521 purchased from BioLamina) and infected at confluency with MOI 1 of each virus in 500 μl volume. Inoculum is removed after 6 h of infection at 37 °C and fresh media added. After 24 h infection, supernatant was collected and frozen at −80 °C until plaque assay was performed. At the same time, cells were washed with 1× DPBS (Gibco) before cell lysis in RLT buffer for RNA extraction.

**NanoString analysis.** EPCs were infected as above and RNA extracted using RNeasy Mini Kit (Qiagen). RNA concentration was determined using the Ribo-Green RNA Assay Kit (Invitrogen) and RNA integrity analyzed using the Agilent RNA 6000 Pico Chip (Agilent). RNA of 150 ng were hybridized to the NanoString nCounter Human Immunology and Inflammation v2 Panels (NanoString Technologies). Hybridized samples were immobilized using the nCounter Prep Station (NanoString Technologies) and scanned using the nCounter Digital Analyzer with the high sensitivity protocol. Data was analyzed using the nSolver Analysis Software (NanoString Technologies). Specific genes analyses were done by normalizing counts obtained for the genes to counts for GAPDH. Heat maps were plotted using the Morpheus web program developed by the Broad Institute with the log$_2$ values of fold changes relative to uninfected samples. Each sample was performed with biological triplicates.

**Viral replication after interferon treatment.** Vero cells were seeded at $2 \times 10^4$ cells per well in 96-well tissue culture plates 1 day prior to infection. Cells were infected at multiplicity of infection (MOI) of 0.05 with or without indicated dilutions of recombinant IFN-λ1 (R&D Systems, 1598-IL-025). Supernatant was collected 48 h post infection and RNA extracted using QIAamp Viral RNA Mini Kit (Qiagen) according to manufacturer's instructions. Viral RNA was quantified using TaqMan qPCR method mentioned below. Percent inhibition from IFN treatment was quantified relative to infection without IFN treatment.

**Plaque reduction neutralization test.** On 21 days post inoculation, mice sera were collected via submandibular bleed. Plaque reduction neutralization test (PRNT) was performed as previously described[16] using 40 p.f.u. of H/PF/2013 instead. Plates were fixed and stained[64] 5 days after infection.

**Animal studies.** All animal studies were performed in accordance to protocols approved by the Institutional Animal Care And Use Committee at Singapore Health Services, Singapore (ref no.: 2016/SHS/1197). Type I IFN receptor-deficient Sv/129 (A129) mice purchased from B&K Universal (UK) were housed in a BSL-2 animal facility in Duke-NUS Medical School. Nine- to 15-week-old mice were used in the experiments. Animals were randomly assigned to different groups. Male mice were injected i.p. with viruses diluted in PBS to stated doses (10$^3$ or 10$^4$ p.f.u. in 200 μl). Daily weight measurements were obtained and submandibular bleed was performed on stated days post infection to obtain serum samples. Mice were sacrificed when exhibiting greater than 20% weight loss. For virus infection in pregnancy, 8–10-week-old female A129 mice were housed with adult male A129 mice in same cages (one female with one male) for 4 days. Female mice were infected i.v. with $1 \times 10^3$ p.f.u. of H/PF/2013, DN-1 and DN-2 on day 10 (corresponding to embryo days 6–10). Organs and fetuses were harvested on days of killing and frozen at −80 °C, until homogenization with TissueLyser (Qiagen) in PBS. Serum viral RNA was extracted using QIAamp Viral RNA Mini Kit (Qiagen) according to manufacturer's instructions. RNA from homogenates of brains, testes, spleens, kidneys, placenta, and whole fetuses were extracted using TRIzol LS (Invitrogen). No blinding was done for animal studies.

**Mosquito infection studies.** A. aegypti mosquitoes (colonies established from eggs collected in Singapore) were orally infected with blood containing 10$^6$ p.f.u. per ml of the Paraiba wild-type strain or DN-2 (20 per group) as previously described[38]. Viral genomic RNA was also quantified as previously described[38].

**Quantitative real-time PCR.** RNA from HuH-7, MRC-5, moDCs, mDMs, EPCs, and HUVECs were extracted using RNeasy Mini Kit (Qiagen) and cDNA synthesized using qScript cDNA Synthesis Kit (Quantabio) according to manufacturers' instructions. Quantitative real-time PCR was performed using LightCycler 480 SYBR Green I (Roche) with primer pairs for ZIKV NS5 (forward: 5′- AARTACACATACCARAACAAAGTGGT-3′ and reverse: 5′-TCCRCTCCCYCTYTGGTCTTG-3′) and YF17D (forward: 5′- GCTAATTGAGGTGYATTGG TCTGC-3′ and reverse: 5′-CTGCTAATCGCTCAAMGAACG-3′) as previously described[16,67], and human glyceraldehyde 3-phosphate dehydrogenase (GAPDH) (forward: 5′-GAGTCAACGGATTTGGTCGT-3′ and reverse: 5′-TTGATTTTGG AGGGATCTCG-3′).

RNA from animal organs and fetuses were quantified with the NanoDrop 2000 Spectrophotometer (ThermoFisher) and diluted to 25 ng/μl for qPCR. qPCR to determine viral RNA progeny levels after IFN treatment, and serum and organ viral loads were performed with 4 μl of RNA using qScript One-Step qRT-PCR kit (Quantabio) in 20 μl reactions with ZIKV 1086 and ZIKV 1162c primers with ZIKV 1107-FAM probe (5′-6-FAM- AGCCTACCTTGACAAGCAGTCAGACAC TCAA-3′-BHQ-1 from Integrated DNA Technologies) as previously described[68]. In vitro-transcribed RNA containing the target region for primers and probe set was used to generate a standard curve for quantification of viral RNA copy number. All qPCR reactions were carried out on the LightCycler 480 RT-qPCR system (Roche) and analyzed with LightCycler 480 Software (Roche). Limit of detection is 40 copies of viral RNA per reaction.

**Statistics.** Graphs shown were plotted and analyzed using the GraphPad Prism software. Two-tailed non-parametric Student's t test was performed on all data sets shown unless otherwise stated. Error bars in graphs represent s.d. with * representing $p < 0.05$, ** representing $p < 0.01$, *** representing $p < 0.001$, and **** representing $p < 0.0001$.

**Data availability.** Raw data files for the microarray have been uploaded to ArrayExpress under the accession no. E-MTAB-6192. The complete genome for PF13/251013-18 has been uploaded to GenBank under the accession number MG827392 and the NGS data for the laboratory stock of the virus has been uploaded to Sequence Read Archive (SRA) under the accession no. SRP131290. The authors declare that the other data supporting the findings of this study are available within the article and the Supplementary materials, or are available from the authors upon request.

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

## Acknowledgements

We thank Karin Sundström for the design of primers used in Gibson assembly, Katell Bidet for providing the vector for infectious clone generation, and also Amanda Bifani for help during the manuscript preparation process. We also thank Nicole Wei Wen Tan for providing help in the pregnant mice work. We thank the Duke-NUS Genome Biology Facility for providing NGS and microarray services. J.P. received support from National Medical Research Council of Singapore (NMRC/BNIG/2032/2015 and NMRC/CBRG/0074/2014). C.C. and S.H. are supported in STIIC with grants from National Medical Research Council of Singapore (NMRC/STaR/020/2013, NMRC/MOHIAFCA T2/005/2015, MOHIAFCAT2001, CIRG13nov032, and NMRC MOHIAFCAT1-6003). This study was funded by the National Medical Research Council of Singapore (NMRC/CSA/0060/2014) and internal support from the Duke-NUS Medical School.

## Author contributions

E.E.O. conceived the study, designed the experiments, and supervised the project. S.S.K., K.R.C., E.Z.O., H.C.T., W.C.N., M.T.X.N., E.S.G. and S.L.Z designed and performed the experiments. S.S.K., K.R.C., E.Z.O., W.C.N. and E.S.G. analyzed the data. M.T.X.N. and K.T. provided the cells, technical expertise, and reagents for the EC experiments. S.W. designed and performed the animal experiments with oversight from S.G.V. and assistance from K.W.K.C. J.H.T. and O.M.S. performed NGS sequencing analyses. M.M. and J.P. performed and analyzed the mosquito infection studies. C.C. and S.H. performed the NanoString nCounter assay. S.S.K. and E.E.O. wrote the manuscript. S.W., K.R.C., W.C.N., M.T.X.N, E.S.G., K.W.K.C, J.H.T., O.M.S., C.C. and S.G.V. reviewed the manuscript.

## Additional information

**Competing interests:** The authors declare no competing interests.

