## [Peer Review File · Nature Communications]

Reviewers' comments:

Reviewer #1 (Remarks to the Author):

Overall, the authors have addressed adequately the points I raised in the review. Regarding my 3rd point: I was not questioning the fact that the interaction of flaviviruses (ZIKV) with myeloid cells is important. It is more the choice of the comparator as immunogenicity studies in humans (have to) rely on PMBCs - for the reasons stated by the authors.

Reviewer #2 (Remarks to the Author):

This is a revised manuscript that has been improved by incorporating responses to the three reviewers.

I have to say that I do not see this manuscript directly leading to a live Zika vaccine candidate, rather it provides some interesting approaches and pitfalls (the term concept is used by the authors) to the development of live attenuated vaccines for flaviviruses, in particular for ZIKV. It should be of general interest to the flavivirus vaccine community

Lines 46-50: While I think the analogy with rubella is interesting, care needs to be taken in interpretation as the epidemiology of rubella is different to that of Zika, and the vaccination strategy would likely not be the same (eg route of transmission, target cells of each virus).

Lines 67-85: Based on the previous reviewer's comments, some text here on the similarities (mosquito-borne diseases) and differences between DENV, YFV and ZIKV and what a live attenuated vaccine needs to do to prevent disease (i.e., multiple cell tropisms of ZIKV vs DENV and YFV) may help.

Lines 116-121: Addition of the NGS data helps improve the manuscript but the data in Fig 2 needs expanding. If the virus gains an additional mutation after passage 3 (NS1), I do not think it is stable up to 7 passages as suggested by the authors. This is an important point for vaccine development and needs some comment.

Lines 196-199: The mosquito infection studies are interesting but I believe the infectivity of DN-2 in mosquitoes would still allow transmission. The authors should evaluate papers of mosquito competence of ZIKV strains, in particular the low infectivity titers needed for transmission.

Lines 213-230 and 273-303: The mouse data are interesting and some comparison with other candidate ZIKV live attenuated vaccines would be beneficial, for example the 3'UTR

mutants. The detection of viral RNA in a minority of reproductive tissues needs more discussion and speculation on the significance.

Reply to Reviewers' Comments

Reviewer #3 only provided confidential comments to editors regarding experiments with NHP. We suggest that the revised manuscript discusses the lack of non-human primate experiments.

We thank the reviewer for the suggestion. We have revised the manuscript to indicate the need for non-human primate studies to further test the safety and immunogenicity of ZIKV live attenuated vaccine (LAV) candidates in lines 394-395.

Reviewers' comments:

Reviewer #1 (Remarks to the Author):

Overall, the authors have addressed adequately the points I raised in the review. Regarding my 3rd point: I was not questioning the fact that the interaction of flaviviruses (ZIKV) with myeloid cells is important. It is more the choice of the comparator as immunogenicity studies in humans (have to) rely on PMBCs - for the reasons stated by the authors.

We thank the reviewer for the clarification. We agree with the reviewer that human immunogenicity studies have to rely on PBMCs. In our study, we are proposing an approach to LAV discovery and development for further testing at a pre-clinical stage. As such, we have focused on the use of monocyte-derived dendritic cells, which allow the recapitulation of early immune gene changes that occur soon after immunization and thus provide an indication of the immunogenicity of the ZIKV LAV candidates when compared to successful flavivirus LAVs such as YF17D and PDK53.

Reviewer #2 (Remarks to the Author):

This is a revised manuscript that has been improved by incorporating responses to the three reviewers.

I have to say that I do not see this manuscript directly leading to a live Zika vaccine candidate, rather it provides some interesting approaches and pitfalls (the term concept is used by the authors) to the development of live attenuated vaccines for flaviviruses, in particular for ZIKV. It should be of general interest to the flavivirus vaccine community

We fully agree with the reviewer. Indeed, our findings suggest an approach to screen and assess potential ZIKV LAVs.

1. Lines 46-50: While I think the analogy with rubella is interesting, care needs to be taken in interpretation as the epidemiology of rubella is different to that of Zika, and the vaccination strategy would likely not be the same (eg route of transmission, target cells of each virus).

We thank the reviewer for this comment. Our intent here is to emphasize the role of long-term herd immunity to prevent congenital infection using rubella as an example. We have now amended this sentence to emphasize the fact that we are drawing epidemiological lessons from a successful childhood vaccination program in preventing congenital disease when vaccinated children reach childbearing age. We agree that the basic biology of rubella

infection is different but the approach to developing a vaccination program is largely similar. We believe our amendments made in lines 49-51 now clarifies the similarity and differences between rubella and Zika.

2. Lines 67-85: Based on the previous reviewer's comments, some text here on the similarities (mosquito-borne diseases) and differences between DENV, YFV and ZIKV and what a live attenuated vaccine needs to do to prevent disease (i.e., multiple cell tropisms of ZIKV vs DENV and YFV) may help.

We thank the reviewer for the suggestion and have incorporated this discussion in lines 384-395. We have also taken the opportunity to emphasize key lessons that are being learnt from the current dengue vaccine as well as observations from the inactivated JE and TBEV vaccines^{1,2} that collectively support the notion that long term flaviviral vaccine-mediated immunity likely requires an LAV.

3. Lines 116-121: Addition of the NGS data helps improve the manuscript but the data in Fig 2 needs expanding. If the virus gains an additional mutation after passage 3 (NS1), I do not think it is stable up to 7 passages as suggested by the authors. This is an important point for vaccine development and needs some comment.

We apologize that our phrasing has given the impression that we think DN-2 is stable in Vero cells. We have now clarified this description in the manuscript to emphasize that the A948G mutation is stable up to 7 passages. Although passaging of DN-2 resulted in fewer single nucleotide variants than DN-1, there is a NS1 mutation that emerged at the 4th passage as pointed out by the reviewer. This mutation gave rise to a threonine-to-serine amino acid change in the wing domain of the NS1 protein^{3,4}. This mutation in NS1 is, however, not known to be associated with altered flaviviral fitness. We thus posit that this mutation is likely an adaptation to Vero cells especially since DN-2 was derived through minimal number of in vitro passages. We also suggest that any genomic stability study will need to be carried out in the cell line that is eventually selected to produce DN-2 for clinical evaluation. In contrast, given the purpose of our current study, which is to demonstrate a pathway to select potential ZIKV LAV candidates, we focused our study on DN-2 without the additional NS1 mutation.

4. Lines 196-199: The mosquito infection studies are interesting but I believe the infectivity of DN-2 in mosquitoes would still allow transmission. The authors should evaluate papers of mosquito competence of ZIKV strains, in particular the low infectivity titers needed for transmission.

We thank the reviewer for this criticism and acknowledge the fact that *Aedes aegypti* mosquitoes are highly competent vectors for ZIKV^{5,6}. We have now rephrased in lines 202-210 to frame the interpretation of our findings in a more epidemiologically relevant context.

5. Lines 213-230 and 273-303: The mouse data are interesting and some comparison with other candidate ZIKV live attenuated vaccines would be beneficial, for example the 3'UTR mutants. The detection of viral RNA in a minority of reproductive tissues needs more discussion and speculation on the significance.

We thank the reviewer for the suggestions. We have further discussed the work on the 3'UTR mutants in the Discussions section in lines 384-395. We have now taken the opportunity to

again emphasize the outcome of the work of the 3'UTR mutants^{7,8}, which again underscores the fact that low level or even absence of viremia correlates with poor immunogenicity and hence vaccine efficacy. We have, however, avoided direct comparison on the rates of organ infection as we have used qPCR instead of the less sensitive plaque assay to assess organ infection. Indeed, our rationale is to use the most sensitive method to detect presence of infection rather than to ask if the organ is producing viable and infectious virions that could be further transmitted. The former, we believe, is more important to assess safety in vaccine recipients. Although it is possible to extrapolate qPCR findings into plaque forming unit equivalents, such an analysis will produce highly speculative findings. Instead, we believe that a future head-to-head comparison of the various LAV candidates in various animal models will be needed to truly assess the safety and efficacy of the various vaccine candidates.

References

- 1 World Health Organization. *Tick-borne Encephalitis*, <<http://www.who.int/ith/vaccines/tbev/en/>> (2017).
- 2 World Health Organization. *Japanese encephalitis*, <http://www.who.int/ith/vaccines/japanese_encephalitis/en/> (2017).
- 3 Brown, W. C. *et al.* Extended surface for membrane association in Zika virus NS1 structure. *Nat Struct Mol Biol* **23**, 865-867, doi:10.1038/nsmb.3268 (2016).
- 4 Xu, X. *et al.* Contribution of intertwined loop to membrane association revealed by Zika virus full-length NS1 structure. *EMBO J* **35**, 2170-2178, doi:10.15252/embj.201695290 (2016).
- 5 Kauffman, E. B. & Kramer, L. D. Zika Virus Mosquito Vectors: Competence, Biology, and Vector Control. *J Infect Dis* **216**, S976-S990, doi:10.1093/infdis/jix405 (2017).
- 6 Pompon, J. *et al.* A Zika virus from America is more efficiently transmitted than an Asian virus by *Aedes aegypti* mosquitoes from Asia. *Sci Rep* **7**, 1215, doi:10.1038/s41598-017-01282-6 (2017).
- 7 Shan, C. *et al.* A live-attenuated Zika virus vaccine candidate induces sterilizing immunity in mouse models. *Nat Med* **23**, 763-767, doi:10.1038/nm.4322 (2017).
- 8 Shan, C. *et al.* A single-dose live-attenuated vaccine prevents Zika virus pregnancy transmission and testis damage. *Nat Commun* **8**, 676, doi:10.1038/s41467-017-00737-8 (2017).

REVIEWERS' COMMENTS:

Reviewer #2 (Remarks to the Author):

The authors have addressed all my comments. The manuscript reads very well.

There is one typo. Line 185: "theses difference" should be these differences"

Address of reviewer comments

Reviewer #2 (Remarks to the Author):

The authors have addressed all my comments. The manuscript reads very well.

There is one typo. Line 185: "theses difference" should be these differences"

Thank you for the favorable review and for pointing out this mistake! The manuscript has been corrected.